# IMPROVE CODE GENERATION WITH FEEDBACK

## ABSTRACT

As advancements in Large Language Models (LLMs) continue to accelerate, an increasing number of researchers are exploring the potential of these models to assist in everyday tasks. Despite their remarkable achievements in various downstream applications, several challenges must be addressed. This paper delves into applying LLMs in coding tasks, such as ChatGPT and LLama. Initial observations suggest that directly employing these LLMs does not yield optimal results. However, we have identified that LLMs demonstrate enhanced performance when given appropriate feedback. This includes providing information on the accuracy of the code generated, supplying test cases relevant to the task, and indicating the correct or incorrect outputs for these test cases. Furthermore, we have developed an innovative architecture miming human debugging. This approach supplies local variable information to the LLM while executing the generated code. Our architecture facilitates providing feedback to the LLM and simulates the human debugging experience, thereby significantly improving the LLM's code generation capabilities. Utilizing our proposed architecture, our model surpasses the current benchmarks of state-of-the-art models in the MBPP and Humaneval datasets. We also present comprehensive analyses and ablation studies to substantiate the efficacy of our methods. These findings open new avenues for enhancing the utility of LLMs in coding tasks, offering a more interactive and practical approach to leveraging these advanced technologies.

## 1 INTRODUCTION

As Large Language Models (LLMs) continue to advance, their integration into various aspects of daily work has become increasingly prevalent. For instance, LLMs have been explored in the healthcare sector to offer insightful analyses of health-related data (Cascella et al. (2023)). Their application in education is also being researched to enhance learning and teaching processes. While these efforts mark significant strides in applying LLMs to improve daily life, their potential applications still need to be fully realized.

One area of recent focus is the use of LLMs in programming, specifically in code writing, as indicated in LEVER (Ni et al. (2023)). However, employing LLMs like ChatGPT for direct coding tasks often yields suboptimal results. A notable challenge is the correction of inaccuracies in the code generated by these models. Several methods have been attempted to address this issue. For example, LEVER (Ni et al. (2023)) generates multiple code variations and employs an executor to select the most effective solution. Meanwhile, LATS (Zhou et al. (2023)) introduces a combination of tree search and an evaluation module to ascertain the most suitable code output. Despite these innovations, a critical gap still needs to be in rectifying errors in the code produced by LLMs.

To understand why LLM perform poor in code generation, we used GPT-3.5-turbo to generate test cases for the MBPP dataset and identified several common issues contributing to poor performance. First, incorrect function and variable names: the MBPP dataset typically provides a problem description like "Write a function to find the longest chain which can be formed from the given set of pairs," but LLMs cannot accurately deduce the true function and variable names required for the test cases. Second, incorrect data structures: for example, given a problem like "Write a function to find the similar elements from the given two tuple lists," the correct solution may need a tuple, but LLMs might return a set, which fails the test cases. Finally, logic errors in the generated code: these are particularly challenging to address and are a common issue with LLM-generated solutions.

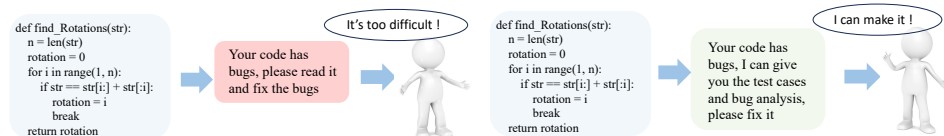

Figure 1: When there are bugs in the code. Only tell a human that there are bugs; it's difficult for humans to fix them. When providing test cases and bug analysis, it's easier for humans to fix the bug

To solve this problem, we thought about how humans debug code. In human programming practices, debugging is an iterative process involving not only identifying that an error exists but also understanding where and why it occurs by analyzing intermediate variables and logic flow during code execution. This detailed feedback is crucial for correcting code and improving its accuracy.

Motivated by this observation, we propose a novel architecture that simulates the human debugging process within LLMs. Our approach provides detailed feedback to the LLM, including specific bug locations, logical discrepancies, and the state of intermediate variables during code execution. By doing so, we enable the LLM to iteratively refine and correct its code generation, significantly enhancing its ability to produce accurate and reliable code.

Our key contributions can be summarized as follows:

- **Identification of Limitations**: We identify the limitations of existing LLM-based code generation methods in correcting code inaccuracies and propose a novel method that emulates the human debugging process.

- **Human-Like Debugging Architecture**: We develop an innovative architecture that supplies detailed feedback to the LLM, including intermediate variable states, mirroring how human programmers debug code by analyzing execution flow and logic errors.

- **State-of-the-Art Performance**: We demonstrate through extensive experiments on the MBPP and HumanEval datasets that our method achieves state-of-the-art performance, surpassing existing models by up to 7% in Pass@1 accuracy. Our ablation studies further validate the effectiveness of our approach.

By integrating human-like debugging processes into LLM code generation, we offer a new, interactive method that enhances the practical utility of LLMs in coding tasks, paving the way for more accurate and efficient code generation in real-world applications.

## 2 RELATED WORKS

### 2.1 ADVANCEMENTS IN LARGE LANGUAGE MODELS (LLM)

Recent advancements in Large Language Models (LLM) have been remarkable, driven by the expansion of training data and substantial computational resources. Pioneering models such as BERT (Devlin et al. (2018)) and GPT-2 (Radford et al. (2019)) laid the groundwork in this domain. A significant milestone was the introduction of GPT-3 in June 2020, which, with its 175 billion parameters, began to significantly impact everyday life. This sparked a trend where numerous entities, including companies and researchers, started developing their own LLMs. Examples include LLaMA (Touvron et al. (2023)), Chinchilla (Hoffmann et al. (2022)), ERNIE (Wang et al. (2021)), and BLOOM (Workshop et al. (2022)). These models have shown exceptional performance in various applications such as chatting, question-answering, and code generation. Moreover, models like InstructGPT (Ouyang et al. (2022)) have enhanced human-model interaction by aligning LLMs with human intent, allowing them to address a broad spectrum of queries. The emergence of open-source LLMs such as Vicuna (Peng et al. (2023)) and Alpaca (Taori et al. (2023)) based on LLaMA further demonstrates the potential of these models.

Figure 2: The pipeline of our model. We start with a task and use LLM to generate an initial code; then, we use our code executor to evaluate the generated code using the test cases. If there are bugs in the generated code, we use our debug module to track the intermediate variables and other information. Based on the collected information, our architecture will try to provide a reason and solution to correct the bugs. Then the LLM can start the next iteraion.

## 2.2 EMPLOYING LLM AS AN AGENT

As LLMs have evolved, efforts have been made to utilize them as agents in everyday scenarios. In this context, an 'agent' refers to an entity where we input information such as objectives, environment, and feedback, allowing the LLM to assist in decision-making. Starting with the concept of Chain-of-Thought (Wei et al. (2022)) and Tree-of-Thought (Yao et al. (2023)), researchers have been teaching LLMs to approach goals incrementally, similar to human processes. In ReAct (Yao et al. (2022)), there is a shift towards guiding LLMs to generate reasons in addition to actions. Additionally, recent studies (Chang et al. (2023),Hu et al. (2023)) are exploring the use of reinforcement learning to provide feedback to LLMs, enhancing their performance. The application of LLMs as agents spans numerous fields. For instance, they have been used in gaming (Qiao et al. (2023),Akoury et al. (2023)), customer service (Cai et al. (2023), Soni (2023)), and code generation (Ni et al. (2023), Zhou et al. (2023)). The expansion of LLMs continues to open up new possibilities for their application in various domains.

## 2.3 CODE GENERATION

The task of generating code based on given tasks has a long history, with earlier methods being predominantly rule-based, as seen in works done by Woods (Woods (1973)) in 1973. However, with the advent of deep learning, there has been a shift towards leveraging these techniques for code generation, exemplified by Xiao (Xiao et al. (2016)). Recently, LLMs have shown remarkable efficacy in this field, leading to numerous innovations aimed at refining LLM-generated outputs. Some recent works (Ni et al. (2023), Zhou et al. (2023)) have experimented with adding an executor to run the generated code and select the best option from multiple outputs. Concurrently, approaches like ReAct (Yao et al. (2022)) focus on providing appropriate feedback based on the generated code and its test cases. Despite these advancements, LLMs face challenges correcting errors in their generated code. In our research, we simulate the human debugging process and have achieved significant progress compared to prior methodologies.

## 3 ARCHITECTURE

### 3.1 ARCHITECTURE OVERVIEW

In our study, we aim to replicate the human approach to identifying and rectifying coding errors. Typically, a programmer detects and corrects bugs by isolating a failing test case, running the code line by line to find the source of the error, and revising the problematic section until it passes all tests. Drawing from discussions with fellow programmers and personal experience, this multi-step debugging process informs our research.

We use a Large Language Model (LLM) to mimic this human debugging process. Recognizing the various stages of human code writing, we divided the task into multiple phases, each simulated by the LLM. The whole architecture can be seen in Fig2 The first phase involves the code generator, which replicates the initial code writing. The second phase introduces a module that simulates human testing and evaluation of the code's correctness through a code executor.

Given the extensive use of test cases in practical applications, we designed a pseudo test case generator to provide numerous test scenarios for code assessment. Our architecture's fourth component is

the debug module, which examines intermediate variables during execution to locate errors, similar to how a human would. The fifth phase involves bug correction. Once a bug is identified through a test case, our feedback module channels this information back to the code generator to facilitate code revision, mirroring the iterative debugging process humans use.

In summary, our architecture includes five key components: an LLM as the main code generator, a pseudo test case generator for diverse test scenarios, an executor to validate the generated code and monitor intermediate variables, a debug module for error analysis, and a feedback module that uses execution results to guide code regeneration. This holistic approach closely mimics the iterative process of human code debugging and refinement.

### 3.2 CODE GENERATOR

In the human coding process, when a programmer is presented with a specific task—such as "Write a function to calculate the volume of a cube given its side length"—they begin by understanding the problem and planning how to implement a solution. They might start by defining the function signature, choosing appropriate function and parameter names, and writing an initial version of the code that implements the basic logic.

Similarly, in our architecture, the code generation process starts with inputting the coding task into the LLM, which acts as the programmer. We provide the LLM with predefined function names and input parameters to mimic how a programmer would set up their code structure, reducing simple errors like incorrect function names or mismatched parameters. The LLM then generates an initial code template that includes the function definition and preliminary implementation.

This template is analogous to a human programmer's first draft of code. At this stage, while we obtain a functional code output, its correctness is not yet assured, nor can we identify potential bugs within the code. Just as a human programmer would proceed to test and refine their code, we rely on the subsequent components of our architecture to validate and improve the generated code. This mirrors the initial step in human coding, where the programmer writes code based on their understanding before entering the testing and debugging phases.

### 3.3 CODE EXECUTOR

In the human coding process, after writing the initial code, a programmer typically runs the code to test its functionality. They execute the code with various test cases to see if it produces the expected results. Importantly, when errors occur, programmers don't just look at the final output; they also examine the intermediate states of the program—such as variable values at different points in execution—to identify where the code is deviating from expected behavior.

Our architecture simulates this aspect of human debugging through the code executor component. While previous methodologies may only check for correctness of the output, our executor not only validates whether the generated code produces the correct results but also meticulously tracks changes in intermediate variables during execution. This is akin to how a programmer might use debugging tools or insert print statements to monitor variable states.

Specifically, we implement mechanisms within the executor to record the dynamics of intermediate variables throughout the code execution process. Upon completion of the test cases, the executor first evaluates if the generated code satisfies all tests. If the code passes, it is deemed correct. If not, the executor compiles a comprehensive report—including the failed test cases, their expected outputs, the actual outputs from the generated code, and the state of intermediate variables. This detailed information mirrors what a human programmer would observe and analyze when debugging their code, providing valuable insights for the next debugging steps.

### 3.4 PSEUDO TEST CASE GENERATOR

In human coding practices, programmers often create their own test cases beyond the examples provided, especially when they anticipate edge cases or wish to thoroughly validate their code. This involves thinking critically about different input scenarios that could potentially cause the code to fail, such as boundary values, unusual inputs, or invalid data.

Our architecture emulates this aspect of human debugging through the pseudo test case generator. Recognizing that standard datasets may lack sufficient test cases to fully challenge the code, we use an LLM to generate additional, diverse test scenarios. Just as a programmer might devise various inputs to test their code's robustness, the LLM generates a range of test cases that may include edge cases or uncommon inputs.

While these LLM-generated test cases may not all be perfect, they serve a crucial role in simulating the human approach to uncover hidden bugs. By incorporating this pseudo-test case generator, we significantly enhance the robustness of our testing process. We execute these test cases using the code produced by our architecture, and any failures encountered during these tests are meticulously recorded. This approach mirrors the comprehensive testing methodology a human programmer would employ to ensure the reliability and correctness of their code.

## 3.5 DEBUG MODULE

In the human debugging process, when a programmer's code does not perform as expected, they analyze the failed test cases to understand why the code is not working correctly. This involves checking whether the test cases themselves are valid and then examining the code to identify logical errors or incorrect assumptions. Programmers often inspect variable values at different points in execution, consider alternative code paths, and hypothesize about the root causes of the errors.

Our architecture's debug module simulates this critical step of human debugging. When the code executor reports failures, the debug module uses an LLM to analyze the detailed information collected—including the failed test cases, expected and actual outputs, and intermediate variable states. The first objective is to verify the validity of the test cases, ensuring that any potential errors in the pseudo-test cases are not misleading the debugging process.

Once the test cases are confirmed to be correct, the LLM delves into the code itself, identifying specific sections that may contain bugs. It offers explanations and possible solutions, much like a human programmer would reason about why the code is failing under certain conditions. This involves examining the logic flow, considering the values of variables at different execution points, and pinpointing where the code diverges from expected behavior. By simulating this analytical process, the debug module helps guide the LLM in understanding and correcting the underlying issues in the code.

## 3.6 FEEDBACK MODULE

In human programming, after identifying potential errors and their causes, a programmer consolidates their findings and uses this insight to modify and improve their code. This iterative process involves applying the knowledge gained from debugging to refine the code, testing the new version, and repeating this cycle until the code functions correctly. Programmers also recognize when they are not making progress and may decide to approach the problem differently to avoid endless cycles of unproductive changes.

Our feedback module emulates this crucial step in the human debugging process. After the debug module analyzes the errors and suggests possible fixes, the feedback module compiles this information and presents it back to the code generator. This includes detailed explanations of the errors, the reasoning behind them, and specific suggestions for code modifications.

Upon receiving this feedback, the code generator embarks on an iterative process of refining and adjusting the code. It strives to produce a version of the code that addresses the identified issues and aligns with the expected outcomes of the test cases. To ensure this iterative process remains efficient and does not get trapped in perpetual cycles of regeneration, we establish a cap on the number of attempts allowed for code correction. This limit is crucial for maintaining a balance between thoroughness and efficiency in the debugging process.

By incorporating this cyclical process of feedback and refinement, our architecture mirrors the way human programmers learn from their mistakes, apply critical thinking, and iteratively improve their code until it meets the desired specifications. This balanced methodology ensures that our architecture debugs efficiently and effectively, closely emulating human cognitive processes in coding and debugging.

## 4 EXPERIMENTS

### 4.1 DATASET

In our experiments, we use two different datasets:

**MBPP dataset**: The MBPP dataset contains basic Python programming problems stated in natural language. The dataset contains 974 problems. For every problem, the dataset contains three different test cases. Following the previous methods(Ni et al. (2023)), we use the first test cases as part of the prompt to generate the template containing the function signatures. We use all three test cases during the test, and only when the generated code passes all three test cases do we think the generated code is correct. If the generated code fails in any test case, we will think the generated code is wrong and have bugs.

**HumanEval dataset**: The HumanEval dataset provides 164 comment descriptions of functions paired with a canonical implementation of each function and several input–output pairs that the function should pass. We follow the same evaluation method as the MBPP dataset.

### 4.2 BASELINE AND EVALUATION METRICS

**Evaluation Metrics** We use Pass@k as our evaluation metrics which is the same as previous works (Zhou et al. (2023)Wang et al. (2023)Shinn et al. (2023))

**Baseline** We compare our methods with several different architecture including Chain-of-Thought (Wei et al. (2022)), ReAct (Yao et al. (2022)), etc. To better understand the effectiveness of our methods, we use different Large Language Models, including GPT-3.5-turbo and GPT-4 etc., as the LLM to test our method and other methods.

| method | | HumanEval | MBPP |
|---|---|---|---|
| LLMs(zero shot prompting) | AlphaCode Li et al. (2022) | 17.1 | - |
| | Incoder Fried et al. (2022) | 15.2 | 17.6 |
| | CodeX Brown et al. (2020) | 47.0 | 58.1 |
| | PalmCoder Chowdhery et al. (2023) | 43.9 | 32.3 |
| | StarCoder Li et al. (2023) | 33.6 | 52.7 |
| | Llama-70B Touvron et al. (2023) | 30.5 | 45.4 |
| | Code Llama-7B Touvron et al. (2023) | 33.5 | 41.4 |
| | GPT-3.5-turbo Achiam et al. (2023) | 56.4 | 52.6 |
| | Claude-instance-1 | 31.1 | 26.9 |
| | GPT-4-turbo Achiam et al. (2023) | 58.6 | 64.8 |
| | GPT-4 Achiam et al. (2023) | 66.1 | 69.3 |
| LLM-based optimisation approaches | CoT Wei et al. (2022) | 46.9 | 54.8 |
| | ReAct Yao et al. (2022) | 56.9 | 67.0 |
| | Reflexion Shinn et al. (2023) | 68.1 | 70.0 |
| | ToT Yao et al. (2023) | 54.4 | 65.8 |
| | RAP Hao et al. (2023) | 63.1 | 71.4 |
| With GPT-3.5-turbo | Self-Edit Zhang et al. (2023) | 62.2 | 56.4 |
| | Self-Planing Jiang et al. (2023) | 65.2 | 58.6 |
| | Self-debugging Chen et al. (2023) | 61.6 | 60.1 |
| | INTERVENOR Wang et al. (2023) | 75.6 | 69.8 |
| | LATS Zhou et al. (2023) | 83.8 | 81.1 |
| | AgentCoder Huang et al. (2023) | 79.9 | 89.9 |
| | Ours | **88.3** | **90.7** |
| | Reflexion Shinn et al. (2023) | 91.0 | 77.1 |
| | Self-debugging Chen et al. (2023) | - | 80.6 |
| With GPT-4 | MetaGPT Hong et al. (2023) | 85.9 | 87.7 |
| | LATS Zhou et al. (2023) | 94.4 | - |
| | AgentCoder Huang et al. (2023) | 96.3 | 91.8 |
| | Ours | **97.2** | **93.2** |
| With StarCoder | Ours | 64.2 | 69.8 |
| With Claude-instance-1 | Ours | 68.2 | 79.4 |
| With PalmCoder | Ours | 66.7 | 76.4 |
| With Code Llama-7B | Ours | 70.8 | 82.1 |
| With GPT-4-turbo | Ours | 90.4 | 92.7 |

Table 1: Quantitative results of our proposed architecture in HumanEval and MBPP dataset, the best results are highlighted in **bold**.

### 4.3 COMPARISON WITH STATE-OF-THE-ARTS

This section compares our method with several other methods using different LLMs. In table1, we show our result compared with other methods in the MBPP dataset and the HumanEval dataset, and our method achieves State-of-the-art in both datasets.

### 4.4 ABLATION STUDIES

In this section, all experiments are done using GPT-3.5-turbo and tested on the HumanEval dataset.

#### 4.4.1 INFLUENCE OF DIFFERENT LEVEL FEEDBACK

In this part of our study, we evaluate how varying degrees of feedback provided by our feedback module affect its performance. This module is capable of delivering feedback at multiple levels, such as assessing the code's correctness, analyzing specific test cases, and examining the intermediate variables produced during code execution. As illustrated in table

Table 2: Ablation study for different level feedback. The result will improve with more feedback.

| True/False | Instance-wise True/False | Instance wise Feedback | Intermediate Variables | Pass@1 |
|---|---|---|---|---|
| ✗ | ✗ | ✗ | ✗ | 56.4 |
| ✓ | ✗ | ✗ | ✗ | 65.4 |
| ✓ | ✓ | ✗ | ✗ | 76.4 |
| ✓ | ✓ | ✓ | ✗ | 83.5 |
| ✓ | ✓ | ✓ | ✓ | 88.3 |

2 of our report, it becomes evident that the LLM's ability to identify bugs and enhance the overall quality of the final output is significantly improved with more comprehensive feedback.

To elucidate the impact of different feedback levels, we offer some illustrative examples. At the most basic level, feedback might simply indicate whether the code is correct or not, with a prompt like "Your code is **wrong**." Moving to a more detailed level, instance-wise true/false feedback provides specifics about the test case where the code fails, for example, "Your code is **wrong** when the **test case** is . . .". Going a step further, instance-wise feedback includes details about the output, such as "Your code is **wrong**, when the **test case** is . . . , **your code output** is . . . , **the right output** is . . .". The most detailed level involves feedback on intermediate variables, framed as "Your code is **wrong**, when the **test case** is . . . , **your code output** is . . . , the **right output** is . . . When your code is running, the **intermediate variables** are . . .".

This trend mirrors the human approach to debugging. Just as a programmer equipped with more information can more easily locate and rectify bugs in the code, the LLM's performance in identifying and correcting errors is similarly enhanced with richer feedback. In contrast, limited information can make the debugging process more challenging and less efficient. Our findings reinforce the notion that the depth and detail of feedback are crucial in effectively guiding both human and machine learning processes in code debugging.

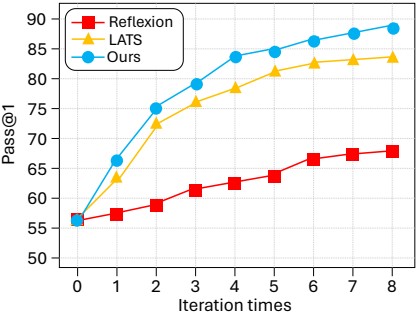

Figure 3: Ablation Study for iteration times. With the increase in iteration times, the result will increase and remain similar. We also compare the influence of iteration in Relexion and LATS. Notice that in the first several iterations, our method have significant improvement compared to Reflexion.

#### 4.4.2 INFLUENCE OF THE MAX ITERATION NUMBER

After receiving feedback, our code generator will generate a new code. However, we cannot guarantee that our method will always provide a correct code. Hence, we must set a max iteration number to avoid our method being stuck in a loop. As shown in Figure3, we can see that with the number of max iterations increasing, the result will first increase and then remain the same, which indicates that for some challenging problems, it's difficult for LLM to generate a correct code. But for some

median difficulty code tasks, giving LLM several chances will help it generate the right code. We also compare the performance of LATS and Reflexion when the iteration numbers increase.

As shown in Fig3, LATS and our model will have an obvious performance improvement in the first several iterations, while the Reflection model does not have a very obvious improvement in the first several iterations. Our analysis of this phenomenon is that LATS and our method will provide high-level explainable feedback rather than low-level implicit feedback. The high-level feedback will help the code generator better understand why the previously code-generated code is wrong and help it better correct its bugs.

### 4.4.3 INFLUENCE OF THE TEMPERATURE

In this part of our research, we delve into how varying the temperature setting in a Large Language Model (LLM) impacts its code generation capabilities. The concept of 'temperature' in the context of LLMs relates to the level of randomness or unpredictability in the generated text. A lower temperature setting results in outputs that are more focused, coherent, and conservative. This means the model tends to produce safer, more predictable text. On the other hand, a higher temperature setting leads to outputs that are more creative and diverse, but potentially less coherent, as the model takes more risks in its text generation.

Our investigations, as illustrated in Figure4, reveal a notable trend: the performance of code generation initially improves with an increase in temperature but eventually starts to decline as the temperature continues to rise. We hypothesize that this phenomenon occurs due to a balance between creativity and coherence. At very low temperatures, the generated code tends to lack creativity. This conservative approach might limit the model's ability to effectively address and rectify the bugs present in the original code. Conversely, at extremely high temperatures, the code produced by the model becomes overly random. This increased randomness can lead to the generation of code that is not only less coherent but also riddled with an excess of new bugs.

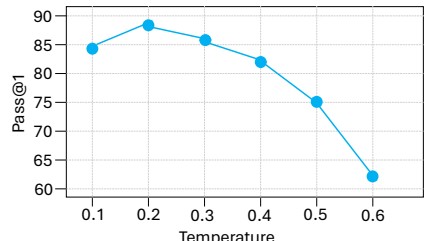

Figure 4: Ablation Study for temperature. Higher temperatures will bring more randomness and hurt the performance of the code generation. Low temperatures will have less creativity and make it difficult to correct the bugs. There is an optimal temperature range where the LLM strikes a balance between creativity and coherence

Therefore, there seems to be an optimal temperature range where the LLM strikes a balance between creativity and coherence, enhancing its performance in code generation. This sweet spot allows the model to be sufficiently innovative to tackle complex coding tasks and fix bugs, while still maintaining a level of predictability and structure that prevents the introduction of too many errors. Understanding this balance is crucial for fine-tuning LLMs in code generation tasks, as it can significantly impact the efficiency and reliability of the output.

### 4.5 USING OUR METHOD TO DEBUG CODE

In this part, we explore an experimental approach distinct from previous methods. While earlier methods concentrated on creating accurate code in response to a coding task, and our methodology has demonstrated superior accuracy, our architecture's capabilities extend beyond mere code generation. Inspired by human debugging processes, we were intrigued to discover if our architecture could also effectively debug existing erroneous code.

We made specific alterations to adapt our architecture for debugging rather than code generation. Our architecture comprises five components, but we omitted the code generator in the initial iteration for the debugging task. Since we already had code containing bugs, generating new code was unnecessary. The other components, however, remained unchanged from their roles in the code generation process.

The method for acquiring flawed code involved using the HumanEval and MBPP datasets. We employed various Large Language Models (LLMs), including GPT-3.5, GPT-4, and LLama, to create code straightforwardly without employing special techniques. The generated code was then tested using the provided test cases, and we collected all code samples that failed these tests. Due to the absence of specialized strategies in code creation, the errors in these samples varied widely, including incorrect function names, input values, logical errors.

In the debugging phase, our code executor gathered more comprehensive information than in its previous use to account for the diverse nature of the bugs. This information included specifics like TypeError and AttributeError. The debug module, therefore, had a broader scope of errors to address and correct. Once these initial errors were rectified, the module shifted to identifying and fixing any logical flaws in the code, using the test cases as a guide, akin to the process in our code generation experiments.

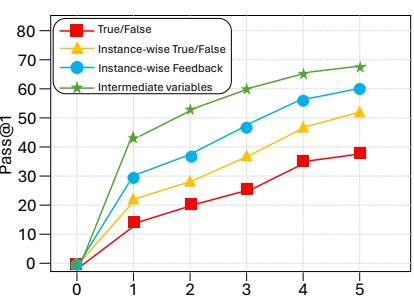

Post-debugging, the original flawed code and the feedback from our debugging process were fed back to the code generator. This step was crucial in generating a corrected version of the code based on the erroneous version and the feedback provided. We then evaluated this newly generated code against all test cases to determine its accuracy. Through this comprehensive process, we aimed not only to identify but also to correct a wide range of coding errors, thereby evaluating the effectiveness of our architecture in a debugging context, an area previously unexplored in our research.

Figure 5: Result of the Accuracy after debugging by our architecture. With more feedback, the accuracy will be higher after debugging. This is intuitive, just like humans, provided more information will help humans better locate and correct the bugs.

To provide a full understanding of the effectiveness of our different levels of feedback, we experimented with different levels of feedback, which have similar settings in our first ablation study. The result can be seen in Fig5

## 5 CONCLUSION

This paper presents a novel architecture that enhances code generation and debugging using Large Language Models (LLMs) by explicitly simulating the human debugging process. Our approach addresses key limitations in current LLM applications, particularly in identifying and fixing coding errors. The architecture's essential components—the code generator, advanced code executor, pseudo test case generator, and innovative feedback module—collaboratively mirror the step-by-step methods a human programmer employs during debugging. This emulation significantly improves the accuracy and efficiency of LLM-generated code.

The advanced code executor goes beyond merely checking code correctness; it meticulously tracks changes in intermediate variables during execution, providing detailed feedback akin to a programmer's observational insights. The pseudo test case generator augments standard datasets like MBPP and HumanEval by creating diverse and comprehensive test scenarios, enhancing the robustness and depth of the debugging process.

Crucially, the debug module identifies error sources in the generated code or test cases, offering detailed insights and potential solutions, much like a human programmer diagnosing and resolving issues. This comprehensive and iterative process ensures efficient and thorough code generation and debugging.

Our architecture achieves state-of-the-art performance on benchmark datasets, demonstrating its effectiveness. By simulating human debugging and providing detailed feedback, we have advanced automated code generation and debugging. These insights will guide the development of more sophisticated and user-friendly coding tools, broadening the applications of LLMs in real-world programming.

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

# 6 APPENDIX

## 6.1 IMPLEMENTATION DETAILS

We use GPT-3.5-turbo for most of our experiments. We set the temperature (influence the randomness of the generated text) to 0.2, and the maximum generation length to 2048.

## 6.2 PROMPTS FOR DIFFERENT LEVEL FEEDBACK

In this section, we provide specific examples for different levels of feedback.

### 6.2.1 TRUE/FALSE

**Q**:  Write a python function to find the minimum number of rotations required to get the same string.
**A**: def find_Rotations(str):
      n = len(str)
      rotation = 0
      for i in range(1, n):
        if str == str[i:] + str[:i]:
          rotation = i
          break
    return rotation
**Q**:  Your generated code is wrong, please fix it.
**A**: def find_Rotations(str):
      n = len(str)
      rotation = 0
      for i in range(1, n):
        if str == str[i:] + str[:i]:
          rotation = i
          break

### 6.2.2  INSTANCE-WISE TRUE/FALSE

> **Q**: Write a python function to find the minimum number of rotations required to get the same string.
> **A**: def find_Rotations(str):
>     n = len(str)
>     rotation = 0
>     for i in range(1, n):
>       if str == str[i:] + str[:i]:
>         rotation = i
>         break
>   return rotation
> **Q**: Your generated code is wrong when test case is 'ab', please fix it.
> **A**: def find_Rotations(str):
>     n = len(str)
>     rotation = 0
>     for i in range(1, n):
>       if str == str[i:] + str[:i]:
>         rotation = i
>         break

### 6.2.3 INSTANCE WISE FEEDBACK

### 6.2.4 INTERMEDIATE VARIABLES

**Q**: Write a python function to find the minimum number of rotations required to get the same string.

**A**: def find_Rotations(str):
    n = len(str)
    rotation = 0
    for i in range(1, n):
      if str == str[i:] + str[:i]:
        rotation = i
        break
  return rotation

**Q**: Your generated code is wrong when test case is 'ab'. Your code output of the test case is 0, the correct output of the test case is 2, please fix it.

**A**: def find_Rotations(str):
    n = len(str)
    rotation = 0
    for i in range(1, n):
      if str == str[i:] + str[:i]:
        rotation = i
        break

### 6.3    MORE EXAMPLES

**Q**: Write a python function to find the minimum number of rotations required to get the same string.

**A**: 
```python
def find_Rotations(str):
    n = len(str)
    rotation = 0
    for i in range(1, n):
        if str == str[i:] + str[:i]:
            rotation = i
            break
    return rotation
```

**Q**: Your generated code is wrong when test case is 'ab'. Your code output of the test case is 0, the correct output of the test case is 2, the local variables when running your code are {'str': 'ab', 'n': 2, 'rotation': 0, 'i': 1}. please fix your code.

**A**: 
```python
def find_Rotations(str):
    n = len(str)
    rotation = 0
    for i in range(1, n):
        if str == str[i:] + str[:i]:
            rotation = i
            break
```

**Q**: Write a python function to count the number of squares in a rectangle.

**A**: def count_Squares(m,n):
   return (min(m, n) * (min(m, n) + 1) * (2 * min(m, n) + 1)) // 6

**Q**: Your generated code is wrong when the test case is count_Squares(4,3): Your code output of the test case is 14, the correct output of the test case is 20, the local variables when running your code are {'m': 4, 'n': 3}. You should calculate the number of squares of size i x i that can fit in the rectangle using the formula (m - i + 1) * (n - i + 1). please fix your code.

**A**: def count_Squares(m, n):
   total_squares = 0
   for i in range(1, min(m, n) + 1):
     total_squares += (m - i + 1) * (n - i + 1)
   return total_squares

---

**Q**: Write a function to convert snake case string to camel case string..

**A**: def snake_to_camel(word):
   parts = word.split('_')
   return parts[0] + ''.join(x.title() for x in parts[1:])

**Q**: Your generated code is wrong when the test case is snake_to_camel('python_program'). Your code output of the test case is 'pythonProgram', the correct output of the test case is 'PythonProgram', the local variables when running your code are {'word': 'python_program', 'parts': ['python', 'program']}. You should consider the first word. please fix your code.

**A**: def snake_to_camel(word):
   parts = word.split('_')
   return parts[0].title() + ''.join(x.title() for x in parts[1:])

