# OpenReview forum: "Improve Code Generation with Feedback"
_ICLR.cc/2025/Conference — Submitted to ICLR 2025_

### Official Review · Reviewer_yNvo · 2024-10-17

**Soundness:** 3
**Presentation:** 2
**Contribution:** 3
**Rating:** 5
**Confidence:** 3

**Summary:**

In this paper, the authors begin by conducting an empirical study to highlight the limitations of existing LLM-based code generation methods in debugging incorrect code. Following this, they propose a novel approach that emulates the human debugging process by providing detailed feedback. This feedback guides LLMs to debug source code similarly to human programmers, focusing on analyzing execution flow and identifying logic errors. The evaluation results indicate that the proposed method achieves state-of-the-art performance compared to existing techniques.

**Strengths:**

1. The method achieves SOTA performance compared to the newest baselines. The authors also conduct an ablation study to demonstrate the contribution of each component.

2. The method is easy to understand and can easily integrated into other frameworks.

**Weaknesses:**

The paper overlooks some important baselines, such as MapCoder and LLM Debugger (LDB). Including these could provide a more comprehensive evaluation and strengthen the overall analysis.

The flow of the writing can be enhanced. The paper's main contributions are centered on the innovative pipeline and the human-like debugging component. It would be beneficial for the authors to elaborate on these aspects in Section 3 rather than covering topics in code generation that are already well-discussed. Furthermore, for lines 347-361, directly including examples in the Appendix could enhance the paper’s readability and clarity.

The datasets used for evaluation, such as HumanEval and MBPP, are relatively easy. Evaluating the proposed method on more challenging datasets, such as BigCodeBench and APPS, would bolster the paper's claims and demonstrate the robustness of their approach.

**Questions:**

How does the effectiveness of this architecture hold up when applied to complex datasets like BigCodeBench and APPS?

How did you determine the number of optimization iterations? In Figure 3, you present results for 8 iterations, whereas Figure 5 shows only 5 iterations. Since the number of iterations can influence the evaluation outcomes, how were these iteration counts chosen, and are they consistent with the baselines?



Minor:

Although less effective than their methods, the authors could include baselines such as MapCoder and LDB in the paper for a more comprehensive comparison.

Line 62 appears to be incomplete.

If the authors address my concerns, I will increase my overall score.

---

### Official Review · Reviewer_o89z · 2024-10-26

**Soundness:** 2
**Presentation:** 2
**Contribution:** 2
**Rating:** 3
**Confidence:** 4

**Summary:**

The paper explores the application of Large Language Models (LLMs) like ChatGPT and LLaMA in coding tasks. Initial observations indicate that direct usage is suboptimal; however, performance can be improved with appropriate feedback. The paper introduces an architecture that simulates the manual debugging process, supplying the LLM with local variable information during code execution. This approach improves the LLM's code generation capabilities by integrating feedback.

**Strengths:**

1. This paper proposes enhancing code generation by simulating the human debugging process.
2. This paper develops an architecture to enhance code generation by providing detailed feedback to LLM.

**Weaknesses:**

1. Similar work already exists in this field. The technical contributions of this paper are limited.
2. The introduction of the five components is too vague. The authors need to provide more technical details of the code generator, pseudo test case generator, executor, debug module, and feedback module. It is best to have a workflow figure for the five components.
3. The structure of the paper need improvement. The author should check the basic punctuation, citations, and grammar, such as "Fig2" and "AlphaCode Li et al., 2022". Furthermore, the most important experimental results, Table 1, is not clear and intuitive.

**Questions:**

1. In this paper, is the proposed architecture fully  automated?
2. How does the debug module assess the validity of the pseudo test cases? I have doubts about the feasibility of using an LLM to validate pseudo test cases.
3. Why does the author claim that "The pseudo test case generator augments standard datasets"? If possible, please provide examples of pseudo test cases.

---

### Official Review · Reviewer_9pGD · 2024-11-03

**Soundness:** 2
**Presentation:** 3
**Contribution:** 2
**Rating:** 3
**Confidence:** 5

**Summary:**

This paper introduces a novel approach to improve code generation in Large Language Models (LLMs) by mimicking human debugging practices. The proposed architecture enhances LLM performance by providing comprehensive feedback during code generation, including code accuracy information, relevant test cases, and local variable states during execution. Through extensive experiments on the MBPP and HumanEval datasets, the approach demonstrates significant improvements, achieving up to 7% better Pass@1 accuracy compared to existing state-of-the-art models.

**Strengths:**

- Debugging for code generation is an interesting and important problem.
- The paper is well-written and easy to follow.
- The proposed method is simple and effective.

**Weaknesses:**

There are several major weaknesses in this paper:

1. No comparison is made with existing debugging methods for large language models (LLMs) in the "Related Work" section, which is a significant oversight. In my view, the idea of including variable information to assist debugging has been explored in LDB [1]. However, this paper lacks a comparison with LDB in the experiments. Additionally, there are lots of other debugging methods that should be discussed and compared in the "Related Work" section, such as [1–4] and so on. The authors should provide a more comprehensive review of existing debugging methods for LLMs and compare their approach with these methods.

   [1] Debug like a Human: A Large Language Model Debugger via Verifying Runtime Execution Step by Step. ACL 2024
   [2] Cycle: Learning to Self-Refine Code Generation. OOPSLA 2024
   [3] Coffee: Boost Your Code LLMs by Fixing Bugs with Feedback. arXiv preprint arXiv:2311.07215
   [4] Selfevolve: A Code Evolution Framework via Large Language Models. arXiv preprint arXiv:2306.02907

2. The experimental results of the proposed method are unusual. Since there are 164 problems in the HumanEval dataset, the change in Pass@1 scores should be approximately 0.6% for each correct prediction. In Table 1, AgentCoder achieves a Pass@1 accuracy of 96.3% with GPT-4, indicating that it correctly solved 158 out of 164 problems. However, the proposed method achieves a Pass@1 accuracy of 97.2% with GPT-4, surpassing AgentCoder by 0.9%.

    I calculated that 159/164 equals 96.95%, and 160/164 equals 97.6%. Neither of these values rounds to 97.2%. The authors should provide more details about the experimental results and explain how the proposed method achieves a Pass@1 accuracy of 97.2%. Similar issues also exist in the other results of the proposed method in Tables 1 and 2. What accounts for the inconsistencies with the observed results?

3. The exact setting of the maximum iteration number used in Section 4.3 is not provided, and the analysis in Section 4.3 is insufficient. The authors should include more details regarding the main results of the comparisons between the proposed method and the baseline methods in Section 4.3.
4. There is a lack of comparison with other debugging methods, such as Self-Debugging and LDB, in the debugging experiments in Section 4.5. The authors should compare the proposed method with these other debugging methods to demonstrate its effectiveness.

And a few minor issues:

1. In Section 4.5, the authors propose to collect buggy codes to analyze the debugging ability of the proposed method. However, they do not provide any statistics about the buggy codes, such as the number of codes collected and the distribution of bug types. The authors should provide more details about the collected dataset.
2. In Figure 4, the authors analyze the performance of the proposed method with different temperature settings ranging from 0.1 to 0.6. I suggest that the authors also provide results for greedy decoding (T=0).
3. In Table 1, the authors should use \citep instead of \citet to change "AgentCoder Huang et al. (2023)" to "AgentCoder (Huang et al. 2023)," which will improve the visual effect of the table. Additionally, there are several other misused citation formats in the paper that should be corrected. I recommend that the authors carefully check the citation format throughout the paper.
4. The current appendix is somewhat disorganized and difficult to read. The authors should ensure that it is neat and clear.

**Questions:**

Please address the concerns in the "Weaknesses" section.

---

### Official Review · Reviewer_5tPE · 2024-11-04

**Soundness:** 1
**Presentation:** 1
**Contribution:** 1
**Rating:** 1
**Confidence:** 5

**Summary:**

This paper presents a code generation method that mimics the human debugging process to improve the initial code generated by an LLM. The proposed method logs the runtime values of variables in the initial code and provides these runtime values along with other information, such as the failed test cases and expected outputs, to the LLM for debugging. It also generates pseudo test cases to facilitate the testing and debugging process. Given the feedback, this method iteratively prompts the LLM to refine the generated code until it reaches a max iteration limit or until the generated code passes all test cases. The authors evaluated the proposed approach on HumanEval and MBPP. They found that it outperformed existing methods by up to 7%. The authors also did additional experiments to investigate the impact of the feedback granularity, the iteration number, and the temperature.

**Strengths:**

1. There has been a lot of interest in LLM-based code generation in the research community recently. This work investigates an interesting and relevant problem.

2. The idea of mimicking human debugging to improve code generation is interesting.

3. In addition to regular experiments on known code generation benchmarks, the authors have done additional experiments to investigate the impact of the feedback granularity, the iteration number, and the temperature.

**Weaknesses:**

1. There is a lack of novelty in this work. The idea of mimicking human debugging has been investigated in LDB (Zhong et al., ACL 2024). LDB also logs the runtime values of variables in the initial code and uses the runtime values as feedback to debug and refine the code. Furthermore, there is no comparison to LDB in the evaluation.

* Li Zhong, Zilong Wang, and Jingbo Shang. Debug like a Human: A Large Language Model Debugger via Verifying Runtime Execution Step by Step. ACL 2024.

2. The description of the proposed method in Section 3 is vague and lacks technical details. The proposed method prompts an LLM for generating pseudo test cases, debugging, and refinement. However, the prompts used for these steps are not provided at all. Besides, since the proposed method logs the runtime values, it is unclear how these values are stored and formatted as part of the feedback sent back to the LLM. For code solutions that involve many variables and intermediate states (e.g., code with a loop), there could be many runtime values. This raises concerns about the soundness and reproducibility of the proposed method given the lack of details.

3. The evaluation has severe rigor issues. Table 1 shows the performance of many existing methods, but the numbers look quite inconsistent with the results of other papers. For example, GPT-3.5 and GPT-4 were reported to have 56.4 and 66.1 pass@1 on HumanEval. However, according to other papers like DeepSeek-Coder, GPT-3.5 and GPT-4 achieved 76.2% and 84.1% on HumanEval. According to the Self-Debug paper (Chen et al., ICLR 2024), Self-Debug achieved 72.8 pass@1 with GPT-3.5 and 80.2 pass@1 with GPT-4 on MBPP when using program traces as part of the feedback. However, this paper only reported 60.1 and 80.6 pass@1 for Self-Debug.

* Chen, Xinyun, et al. Teaching Large Language Models to Self-Debug. ICLR 2024.

4. In the temperature experiment (Section 4.4.3), the authors should repeat the experiments multiple times and report the average value when setting the temperature to a non-zero value since the code generation becomes non-deterministic and can vary significantly with a large temperature value.

5. In the code bugging experiment (Section 4.5), it is unclear how the debugging accuracy is measured.

6. There have been quite a few methods that leverage fine-grained feedback to improve LLMs for code generation in recent years, such as Self-Debug (Chen et al., ICLR 2024), Self-Refine (NeurIPS 2024), and Reflexion (Shinn et al., NeurIPS 2024). More recently, the multi-agent LLM frameworks like AutoCodeRover (Zhang et al., ISSTA 2024). The authors only discussed LEVER in the related work and missed all other related work in this direction. While the authors discussed ReAct, ReAct is not a feedback-driven approach. Instead, it focuses on prompting LLMs to perform internal reasoning and planning.

* Madaan, Aman, et al. Self-refine: Iterative refinement with self-feedback. NeurIPS 2024.

* Shinn, Noah, et al. Reflexion: Language agents with verbal reinforcement learning. NeurIPS 2024.

* Zhang, Yuntong, et al. Autocoderover: Autonomous program improvement. ISSTA 2024.

7. There are many grammar issues and typos in the paper.

**Questions:**

1. How would you compare the proposed method with LDB?

2. Can you provide the prompts used in the proposed method and elaborate on the technical details?

3. What are the experiment setup and hyperparameter settings for the methods in Table 1? What specific API versions are used for GPTY-3.5.-turbo and GPT-4?

4. How was the debugging accuracy measured in Section 4.5?

---

### Meta-Review · Area_Chair_NEPz · 2024-12-21

**Metareview:**

This paper proposes a novel approach to improving code generation in Large Language Models (LLMs) by simulating the manual debugging process. However, several key issues were raised by the reviewers. Firstly, the paper ignores relevant prior work in the field, which weakens its contribution and context. Secondly, the overall presentation is lacking—many parts of the paper appear rushed, with unclear explanations and a disorganized structure. These issues make it difficult to assess the proposed method fully. While the idea has potential, the lack of engagement with prior research and poor presentation significantly hinder the paper's quality. The authors are encouraged to revise the manuscript based on the reviewers' feedback and consider resubmitting it to a future venue. With improvements, there is a strong chance for a successful outcome in future submissions.

**Additional Comments On Reviewer Discussion:**

The reviewers have raised several important concerns, including issues related to prior literature and the overall presentation.

---

### Decision · Program_Chairs · 2025-01-22

Reject